# Role of Klotho as a Modulator of Oxidative Stress Associated with Ovarian Tissue Cryopreservation

**DOI:** 10.3390/ijms222413547

**Published:** 2021-12-17

**Authors:** Boram Kim, Hyunho Yoon, Tak Kim, Sanghoon Lee

**Affiliations:** 1Department of Obstetrics and Gynecology, Korea University College of Medicine, Seoul 02841, Korea; evelynk1219@gmail.com (B.K.); tkim@korea.ac.kr (T.K.); 2Department of Medical and Biological Sciences, The Catholic University of Korea, Bucheon 14662, Korea; hyoon@catholic.ac.kr

**Keywords:** cryopreservation, N-acetylcysteine, klotho, antioxidant

## Abstract

Ovarian tissue cryopreservation is the only option for preserving fertility in adult and prepubertal cancer patients who require immediate chemotherapy or do not want ovarian stimulation. However, whether ovarian tissue cryopreservation can ameliorate follicular damage and inhibit the production of reactive oxygen species in cryopreserved ovarian tissue remains unclear. Oxidative stress is caused by several factors, such as UV exposure, obesity, age, oxygen, and cryopreservation, which affect many of the physiological processes involved in reproduction, from maturation to fertilization, embryonic development, and pregnancy. Here, freezing and thawing solutions were pre-treated with N-acetylcysteine (NAC) and klotho protein upon the freezing of ovarian tissue. While both NAC and klotho protein suppressed DNA fragmentation by scavenging reactive oxygen species, NAC induced apoptosis and tissue damage in mouse ovarian tissue. Klotho protein inhibited NAC-induced apoptosis and restored cellular tissue damage, suggesting that klotho protein may be an effective antioxidant for the cryopreservation of ovarian tissue.

## 1. Introduction

Preserving fertility in women of childbearing potential receiving cancer treatment is a global concern [1]. Early referral for fertility preservation in these patients is necessary because the chances of assisted reproductive success are reduced after cancer treatment [2]. However, the cryopreservation process entails some limitations, including the oxidative stress caused by several external environmental factors that can severely damage cellular structures [3]. Furthermore, cryopreservation at extremely low temperatures inactivates cellular metabolism, which can lead to cellular damage during freezing and thawing. Therefore, from the perspective of ovarian tissue transplantation, slow freezing is more suitable than vitrification for tissue cryopreservation [4]. However, it also increases reactive oxygen species (ROS) levels in tissues that have undergone slow freezing [5].

Oxidative stress is caused by an imbalance between reactive oxygen species (ROS) formation and cellular defense mechanisms [3]. Moderate concentrations of ROS are required for the signaling processes involved in growth and apoptosis protection, while elevated ROS levels caused by an imbalance in antioxidant defenses can lead to oxidative stress associated with aging and many diseases [6,7]. Moreover, high levels of ROS cause the modification of macromolecules, mainly lipids, proteins, and nucleic acids, resulting in significant damage to cellular structures [8]. Excessive ROS levels can affect various physiological functions of the female reproductive system, affecting the overall fertility of women [9]. Excessive ROS during ovarian tissue cryopreservation negatively affects embryo quality and developmental potential due to decreased ovarian quality and increased DNA fragmentation [10,11]. In particular, developing embryos are susceptible to oxidative stress due to cellular structural damage and a lack of defense mechanisms.

Antioxidants protect oocytes from ROS-induced cellular damage. N-acetylcysteine (NAC), a synthetic precursor of intracellular cysteine and glutathione, is an important antioxidant [12]. Park et al. demonstrated that NAC inhibited cell inflammation and apoptosis by scavenging high-glucose-induced ROS [13], suggesting that NAC may be used as an anti-apoptotic agent. Klotho, a transmembrane protein, regulates the organism’s sensitivity to insulin, which is involved in aging [14]. One notable feature associated with klotho-overexpressing cells and tissues is their relatively low oxidation state, while in klotho-deficient systems, the oxidative stress levels are much higher [15,16], suggesting that klotho activity reveals pathways and crosstalk that control oxidative stress levels.

The effect of klotho protein for cryopreservation on cellular changes in ovarian tissue remains unclear. Here, we evaluated the solution treated with N-acetylcysteine and klotho protein during the freezing and thawing of mouse ovarian tissue. Immunohistochemistry (IHC) and immunoblotting data to evaluate oxidative stress-induced DNA damage and apoptosis revealed that klotho protein may be a promising preservative for ovarian tissue cryopreservation by modulating oxidative stress.

## 2. Results

### 2.1. Morphological Changes after Cryopreservation of Ovarian Tissue

To investigate the effect of ovarian cryopreservation on tissue, mouse ovarian tissue was thawed after 4 weeks of cryopreservation. NAC dose (5 mM) was determined by a study on ovarian tissue cryopreservation for patients with premature ovary insufficiency caused by cancer treatment [17]. Recombinant klotho protein dose (0.4 μg/mL) was determined by endotoxin level testing. The morphology of the primordial follicles in each group was evaluated using transmission electron microscopy (TEM). Because primordial follicles are essential indicators of female fertility, we investigated the numbers. As a result, there was a slight decrease in the frozen-tissue group, but it was not statistically significant (Appendix A). Hematoxylin and eosin (H&E) staining barely showed basal layers around the primordial follicle in the mouse ovarian tissues and slow-frozen tissue treated with PBS, NAC, klotho, and their combination (Figure 1—top). Thus, TEM was performed to clearly show the basal layer. Under TEM, the basal layer around the primordial follicle clearly separated the inside from the outside in the control group. While slow-frozen tissue with PBS showed a damaged basal layer around the primordial follicle, mouse ovarian tissue pre-treated with NAC, klotho, and their combination showed a restored primordial follicle basal layer (Figure 1—middle). Interestingly, high-resolution TEM showed that the mitochondrial structure was disrupted in slowly frozen tissue, implying that mitochondria may not survive under slow-freezing conditions. Treatment with NAC, klotho protein, and their combination restored the mitochondrial structure in slow-frozen mouse ovarian tissue (Figure 1—bottom). These data suggest that NAC and klotho could protect against morphological changes from the cryopreservation of ovarian tissue by the slow-freezing method.

### 2.2. Ovarian DNA Damage Due to Oxidative Stress after Cryopreservation

Oxidative stress during cryopreservation increases sperm DNA damage, leading to DNA fragmentation [18,19,20]. To determine whether cryopreservation-induced oxidative stress is mediated by ovarian DNA damage, immunochemical staining was performed with γ-H2AX, a marker for double-stranded breaks (DSBs) caused by DNA damage, and 8-hydroxy-2′-deoxyguanosine (8-OHdG), a biomarker for oxidative stress, which is one of the predominant forms of free-radical-induced oxidative lesions. While slow freezing mouse ovarian tissue markedly increased γ-H2AX compared to that in the control ovarian tissue, treatment with NAC, klotho, and a combination significantly reduced the slow-freezing-induced γ-H2AX (Figure 2A,B). However, despite no statistical significance, the combination treatment slightly induced DSBs compared to slow freezing with single treatment, suggesting that high-dose combination treatment of NAC and klotho may induce cell cytotoxicity, resulting in more double-strand breaks than single treatment. Thus, the concentration that can cause DNA damage should be considered when the NAC and klotho combination treatment is used for slow freezing, which supports that klotho treatment is more effective for cryopreservation than the NAC and klotho combination treatment. In addition, increased 8-OHdG in slow-frozen mouse tissues was inhibited by treatment with NAC, klotho, and their combination (Figure 2A,C). Furthermore, qPCR of *GADD45A*, a growth arrest and DNA-damage-inducible protein, showed that slow-freezing-induced *GADD45A* expression was inhibited by NAC, klotho, and their combination treatment (Figure 2D). These data suggest that the dramatic increase in DNA damage and oxidative stress in the slow-freeze control was regulated by NAC and klotho treatment in ovarian tissue cryopreservation.

### 2.3. Effects of Klotho Protein on Aging by Cryopreservation

In a previous study, we found that cryopreservation is closely associated with the aging process of human ovarian tissue [21]. To test the effect of NAC and klotho on cryopreservation-induced senescence, IHC of p16INK4a, a biological aging marker, was performed under the indicated conditions (Figure 3A,B). Interestingly, p16INK4a levels were significantly higher in NAC-treated ovarian tissues than in slowly frozen control tissues. Conversely, the group treated with klotho protein significantly reduced NAC-induced p16INK4a expression (Figure 3A,B). These data demonstrated that klotho protein could protect against cryopreservation- or NAC-induced senescence in ovarian tissue. To determine the effect of klotho on the stress response, qPCR of *Sirtuin1* (*SIRT1*) involved in cellular regulation and longevity was performed [22]. The data also showed that NAC-induced *SIRT1* expression was significantly reduced by klotho protein under slow-freezing conditions (Figure 3C). These data demonstrated that klotho protein could protect against cryopreservation- or NAC-induced stress in ovarian tissue.

### 2.4. Effects of Klotho Protein on External Environmental Stress and Apoptosis Protection

TEM showed that NAC treatment induced abnormal nuclear architecture in slowly frozen ovarian mouse tissues (Figure 4A—top). Based on these observations, we hypothesized that NAC induces ovarian tissue apoptosis. To determine the effect of NAC and klotho treatment on apoptosis during cryopreservation, IHC was performed with an anti-poly (ADP-ribose) polymerase (PARP) antibody, which is involved in DNA repair in response to environmental stress and serves as a marker for cells undergoing apoptosis (Figure 4A—bottom). While PARP was increased in the slow-frozen group treated with NAC, klotho treatment significantly decreased NAC-mediated PARP in the slow-frozen tissue (Figure 4A,B). Furthermore, immunoblotting analysis of apoptosis-mediating proteins showed that slow cryopreservation and NAC treatment induced apoptosis in cryopreserved ovarian tissue (Figure 4C; Appendix A). However, klotho protein significantly reduced the expression of apoptosis-mediated proteins, suggesting that klotho could be a promising anti-apoptotic agent in slowly frozen ovarian tissue.

## 3. Discussion

In this study, NAC and klotho proteins were evaluated to find cryoprotectants for ovarian tissue cryopreservation. Our results suggest that both NAC and klotho can not only protect against morphological changes from cryopreservation of ovarian tissue, but also significantly reduce DNA damage and oxidative stress induced by the slow-freezing method. However, NAC increased ovarian tissue senescence and apoptosis while klotho decreased those effects caused by cryopreservation or NAC, suggesting that klotho protein may be a promising preservative by regulating ovarian cell aging and apoptosis in ovarian tissue cryopreservation.

Although ovarian cryopreservation is a promising strategy for preserving fertility in women needing cancer treatment [23], there are still limitations to be addressed. ROS produced during cryopreservation can damage lipids, proteins, and nucleic acids. Damage caused by ROS can severely affect the physiological function of cells and reduce the effectiveness of cryopreservation [24]. However, cryoprotectants in ovarian tissue cryopreservation can cause tissue damage by increasing oxidative stress [25]. Therefore, when cryopreservation is performed using antioxidants and cryoprotectants, damage to lipids, proteins, and nucleic acids is reduced. Additionally, osmotic and mechanical damage caused by ice crystals are also reduced, increasing cell viability, motility, and fertility, which improves cryopreservation [24]. However, the use of high concentrations of antioxidants does not necessarily have a positive effect. High concentrations of antioxidants can reportedly negatively affect the cell structure and function by shifting cells from oxidative stress to reductive stress [26]. Moreover, Khor et al. found that cryopreserved sperm with antioxidants may negatively affect the intrinsic immobility mechanism of cells [27]. Therefore, the use of antioxidants as ROS scavengers should be investigated further.

There are two methods of cryopreservation of ovarian tissue: slow freezing and vitrification.

Various studies on ovarian tissue freezing methods are being conducted worldwide. In 2004, Donnez et al. reported the first childbirth through the transplantation of human ovarian tissue [28]. Currently, more than 100 children have been born via freeze-thawed ovarian tissue transplantation [29]. Birth continued only when a slow-freezing procedure was used. The advantages of vitrification among cryopreservation methods are the low risk of ice crystal formation, resulting in minor DNA damage, shorter processing times, and inexpensive equipment [30,31]. However, in this study, slow freezing was performed for ovarian tissue cryopreservation to prevent DNA damage, as vitrification is limited to the state of oocytes or embryos [4].

Here, the effects of antioxidants on oxidative stress regulation, aging, DNA fragmentation, and apoptosis by the cryopreservation of ovarian tissue were investigated. NAC-treated samples showed fewer DNA fragments and lower DNA damage. Gamma-H2AX and 8-OHdG were used as indicators of DNA damage in cryopreserved ovarian tissues. However, NAC also increased apoptosis and autophagy induction, even though NAC had a sufficient ROS scavenging effect during cryopreservation. Therefore, further studies were conducted to find a better cryoprotectant that could replace NAC in ovarian tissue cryopreservation.

The discovery of telomerase and the klotho endocrine system represents a novel approach for treating aging and age-related disorders [32]. Persistent oxidative stress is believed to be associated with aging, a response to cellular stress [33], and many studies have reported that klotho protein may act as an anti-aging hormone in mammals [15]. Furthermore, klotho acts as a humoral factor, reducing apoptosis and cell aging caused by hydrogen peroxide in vascular cells [34]. Based on these findings, we investigated whether klotho can be a cryoprotectant in ovarian tissue during freeze-thaw processes. The number and morphological evaluation of primordial follicles are known indicators of fertility [35]. We showed that klotho protects primitive follicle morphological changes and maintains cell morphology, where the basal layer can clearly be observed through TEM. Age-related p16INK4a staining showed that klotho had anti-aging effects on ovarian tissue cryopreservation. Additionally, klotho protein inhibited NAC-induced apoptosis and autophagy in slowly frozen ovarian tissue.

In cryopreservation, many studies have reported that compounds with known antioxidant properties, such as vitamin C, vitamin E, resveratrol, glutathione, coenzyme Q, and melatonin, are added to cryopreserved cells in association with ROS. However, not all antioxidants contribute equally to their ability to protect cells from freeze-thaw-mediated stress [36]. Supplementation of the cryopreservation medium with antioxidants generally increases the cell viability, indicating that oxidative stress adversely affects cells, such as inducing apoptosis during cryopreservation [37,38]. Caspase activation, which is known to mediate the apoptosis cascade, was also observed in cryopreserved sperm cells [39]. To date, cryopreservation treated with antioxidants has mainly been performed in cells. This study is the first klotho study conducted on the cryopreservation of ovarian tissue. Furthermore, we are currently trying to find a combination of klotho protein and other antioxidants for the ovarian cryopreservation.

In conclusion, adding klotho protein to the conventional slow-freezing method is practical for ovarian tissue aging and follicle survival. It reduces ROS-induced DNA damage and apoptosis, highlighting the reproductive potential of ovarian tissue cryopreservation.

## 4. Material and Methods

### 4.1. Animals and Experimental Design

All animal experiments were approved by the Korea University Institutional Animal Use Committee and were maintained in a pathogen-free state in a barrier facility (KOREA-2020-0136). Female adolescent BALB/C mice (*n* = 30) were used in this study. After euthanasia using carbon dioxide, all ovarian tissues were collected from 6–7-week-old mice. Ovarian tissues from each group were isolated from at least six mice. Sixty ovarian tissue slices were randomly distributed to the control and slow-frozen groups treated with PBS, NAC, klotho protein, and their combinations. Ovarian tissue samples were cryopreserved and thawed by adding the experimental drug to the cryopreservation solution. Tissues were evaluated for primordial follicle morphological changes, double-stranded DNA damage, senescence, and apoptosis after the freeze-thaw process.

### 4.2. Slow Freeze-Thaw Protocol

The slow-freeze and thaw protocols have been previously described [4,21]. Briefly, ovarian tissues were transferred to individual freezing tubes containing 1 mL of medium, and then NAC, klotho protein, and their combination were added to the medium. The cryotubes were cooled in a programmable controlled-rate freezing device (Planer PLC, Middlesex, UK) using a slow-freezing protocol as previously described. It featured cooling from 4.0 °C to −7.0 °C at a rate of −2.0 °C/min, followed by manual seeding and cooling to −40.0 °C at a rate of −0.3 °C/min and then to −140.0 °C at a rate of −10.0 °C/min. Next, the samples were stored at −196.0 °C in liquid nitrogen for 4 weeks. The stored cryotube vials were then transferred to a shaking bath at 37 °C to thaw the vials.

### 4.3. H&E Staining

Ovarian tissue samples from each group were fixed with 4% formaldehyde. The fixed samples were dehydrated with a step-by-step ethanol solution (70–100%) and xylene. After paraffin embedding, sample slices (3 μm thick) were stained with H&E. Stained samples were examined using a microscope (Olympus-BX53, Tokyo, Japan) at 400× magnification.

### 4.4. RNA Isolation and Complementary DNA Synthesis

Total mRNA was extracted from all samples using TRIzol reagent (Thermo Scientific, Waltham, MA, USA) according to the manufacturer’s protocol. The concentration of the extracted mRNA was measured using a NanoDrop spectrophotometer (ND-1000, Thermo Scientific). One microgram of mRNA was synthesized using a complementary DNA kit (Takara, Shiga, Japan) in a total volume of 20 μL at 42 °C for 30 min.

### 4.5. Quantitative Real-Time PCR (qPCR)

qPCR was performed on a CFX96 system (Bio-Rad, Hercules, CA, USA) using SYBR Green Supermix (Cat#: 1708882, BIO-RAD). The qPCR step consisted of an initial denaturation at 95 °C for 10 s, followed by 35 cycles of annealing (55 °C for 30 s) and extension (72 °C for 30 s). Relative gene expression was quantified using the ΔCt method and normalized to GAPDH expression. The primer sequences were as follows: *GADD45a* (forward, 5′-TGGTGACGAACCCACATTCAT; reverse, 5′-ACCCACTGATCCATGTAGCGAC); *SIRT1* (forward, 5′-ACCAAATCGTTACATATTCC; reverse, 5′-CAAGGGTTCTTCTAAACTTG); *GAPDH* (forward, 5′-AGAACATCATCCCTGCATCC; reverse, 5′-TCCACCACCCTGTTGCTGTA).

### 4.6. Transmission Electron Microscopy (TEM)

Mouse ovarian tissues were fixed with 2.5% glutaraldehyde in 0.1 M phosphate buffer at 4 °C overnight. The samples were then fixed with 1% osmium tetroxide, dehydrated, and embedded in Epon mixture (Polybed 812 embedding kit/DMP-30; Polysciences, Inc., Warrington, FL, USA). Sections (1 μm thick) were obtained using a Reichert-Jung Ultracut E ultramicrotome (Leica, Wetzlar, Germany) and stained with toluidine blue. Sections that were 60 nm thick per block were collected on copper grids. These were stained with uranyl acetate/lead citrate, and their morphologies were analyzed using a TEM H-7500 apparatus (Hitachi, Tokyo, Japan) at 80 kV.

### 4.7. Immunohistochemistry (IHC)

Unstained and paraffin-embedded ovarian tissue slides were deparaffinized with xylene and rehydrated with graded ethanol. For antigen retrieval, the slides were treated with Tris-EDTA (pH 9.0; Thermo Scientific) for 30 min at room temperature. The samples were blocked with hydrogen peroxide solution (Cat#: 88597, Sigma-Aldrich, St. Louis, MO, USA) for 10 min. The slides were stained using a Polink-2 Plus HRP Broad Kit (Cat#: D41-18, GBI Labs, Bothell, WA, USA) with DAB (3,3′3diaminobenzidine), according to the manufacturer’s protocol. The slides were incubated at 4 °C overnight in a humid chamber with primary antibodies: γ-H2AX (1:500 dilution; Bethyl Laboratories, Montgomery, AL, USA), 8-OHdG (1:500, Cat#: sc-66036, Santa Cruz, Dallas, TX, USA), PARP (1:400, Cat#: 46D11, Cell Signaling Technology, Danvers, MA, USA), and CDKN2A/p16INK4a (1:100, Cat#: EPR20418, Abcam, Cambridge, MA, USA) in PBS containing 2% BSA. The slides were incubated for 1 h with the following secondary antibodies: goat anti-rabbit IgG (H + L), Alexa Fluor 488 conjugate (1:1000, Invitrogen, Waltham, MA, USA), goat anti-rabbit IgG (H + L), and Alexa Fluor 594 conjugate (1:1000, Invitrogen). The slides were counterstained with Mayer’s hematoxylin (Cat#: HMM125, Scytek, West Logan, WV, USA) and antifade mounting medium with DAPI (Cat#: H-1200, Vector Laboratories, Burlingame, CA, USA).

### 4.8. Western Blot Analysis

Total protein 20 μg was subjected to 8–12% sodium dodecyl sulfate-polyacrylamide gel electrophoresis (SDS-PAGE). The membranes were incubated at 4 °C overnight with the following primary antibodies: β-actin (1:1000, Cat#: 4970, Cell Signaling Technology), anti-cytochrome C (1:1000, Cat#: AB13575, Abcam), anti-BAX (1:1000, Cat#: ab32503, Abcam), FoxO3a (1:1000, Cat#: 12829, Cell Signaling Technology), Phospho-FoxO3a (1:1000, Cat#: 9466, Cell Signaling Technology), SOD (1:1000, Cat#: 13141, Cell Signaling Technology), and GADD45a (1:1000, Cat#: 4632, Cell Signaling Technology). The membranes were incubated for 1 h with the following secondary antibodies: goat anti-rabbit secondary antibody (1:5000 dilution, Cat#: ab6721, Abcam) and goat anti-mouse secondary antibody (1:5000, Cat#: ab6789, Abcam). Protein expression was quantified using the BIO-RAD program. Western blotting was repeated three times.

### 4.9. Statistical Analysis

The results from IHC and immunofluorescence (IF) staining were quantified using ImageJ. Data are expressed as the mean of three independent experiments (±standard deviation) and are presented as tables and bar graphs. The differences among the groups were analyzed by one-way analysis of variance (ANOVA), and a *p*-value < 0.05 was considered statistically significant.

## Figures and Tables

**Figure 1 ijms-22-13547-f001:**
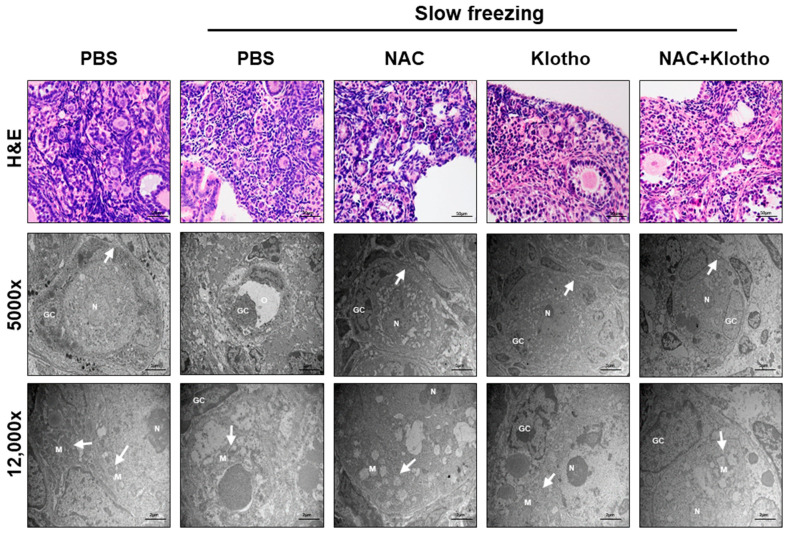
Effect of NAC and klotho proteins on mouse primordial follicle and mitochondrial morphology after slow-freezing cryopreservation. H&E staining was performed to determine the morphological changes in the mouse ovarian tissue after cryopreservation (**top**). Electron micrograph of a mouse primordial follicle showing flattened granulosa cells (GC), nucleus (N), mitochondria (M), and basal lamina (white arrow) (5000× magnification; **middle**). Changes in mitochondrial morphology (white arrow) under the indicated conditions (12,000× magnification; **bottom**). *n* = 2 each group.

**Figure 2 ijms-22-13547-f002:**
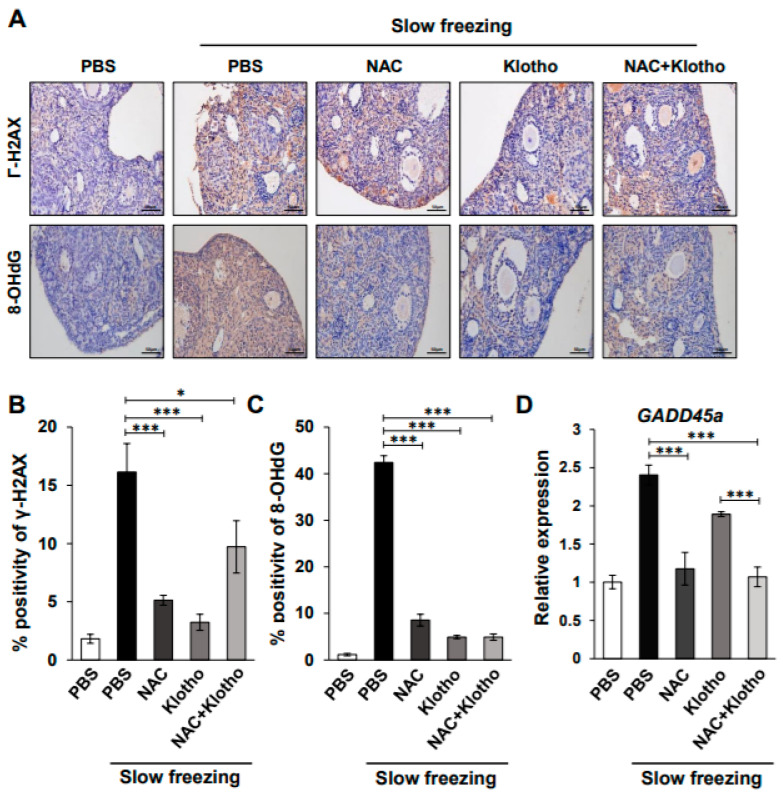
NAC and klotho treatment reduce increased DNA damage and oxidative stress in the slow-freezing method. (**A**) Representative IHC images at 400x magnification showing double-stranded breaks (DSBs) and 8-hydroxy-2′-deoxyguanosine (8-OHdG) in the indicated conditions. Brown indicates DSBs-positive (top) and 8-OHdG-positive in the ovarian tissue (bottom). (**B**,**C**) Graph showing the average percentage of DSBs (**B**) and 8-OHdG (**C**) in the IHC images. (**D**) qPCR results showing relative expression levels of *GADD45a*. All data are shown as mean ± SD. The *p* values were determined by one-way analysis of variance (ANOVA). * *p* < 0.05; *** *p* < 0.0001. *n* = 6 each group. Each experiment was repeated three times.

**Figure 3 ijms-22-13547-f003:**
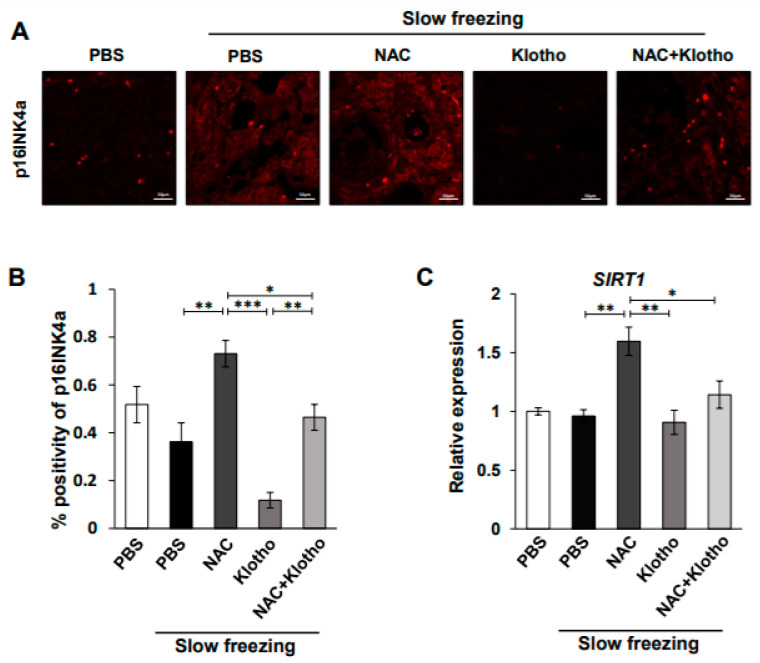
Klotho protein protects NAC-induced senescence in ovarian tissue. (**A**,**B**) Representative IF images at 400× magnification of ovarian tissues with p16INK4a. The positive area for those receptors is shown as red. (**B**) Percentage of p16INK4a-positive cells. (**C**) qPCR of *SIRT1* genes in ovarian tissue after cryopreservation. All data are shown as mean ± SD. The *p* values were determined by one-way ANOVA. * *p* < 0.05; ** *p* < 0.01; *** *p* < 0.001. n = 6 each group. Each experiment was repeated three times.

**Figure 4 ijms-22-13547-f004:**
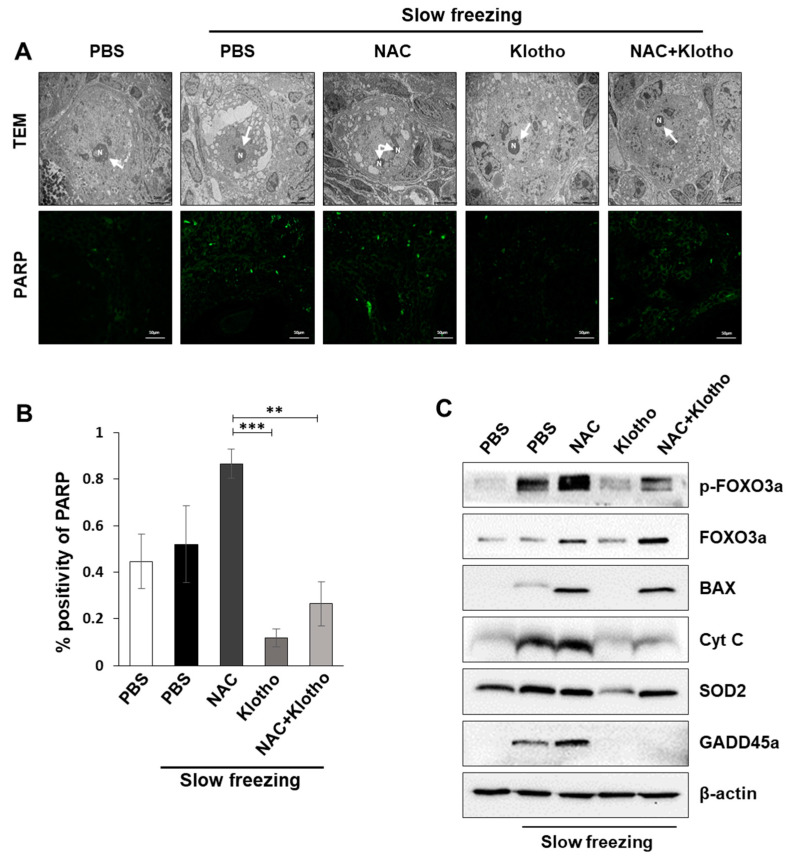
Klotho protein reduces cell death produced by oxidative stress caused by cryopreservation. (**A**) Representative image showing the fragmented nuclear structure observed with a TEM (5000× magnification; top) and IF images of PARP in the indicated conditions (400× magnification; bottom). (**B**) Percentage of PARP-positive cells analyzed by ImageJ. (**C**) Expression of apoptosis-related proteins, including pro-apoptotic protein Bax and anti-apoptotic protein Bcl-2, was detected by immunoblotting. All data are shown as mean ± SD. The *p* values were determined by one-way ANOVA. ** *p* < 0.01; *** *p* < 0.001. *n* = 6 each group. Each experiment was repeated three times.

## Data Availability

All data is included in manuscript figures.

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
