# Peer review of "Role of Klotho as a Modulator of Oxidative Stress Associated with Ovarian Tissue Cryopreservation"

_ijms, 2021, doi:10.3390/ijms222413547_

Round 1
Reviewer 1 Report
The manuscript, “Role of Klotho as a Modulator of Oxidative Stress Associated with Ovarian Tissue Cryopreservation” by Kim and colleagues reports the protective effects of Klotho protein on cryopreservation (by slow freezing) of ovarian tissues. The study is well designed and executed in general, and the claims made by the authors are in most part supported by the data. However, before the manuscript can be considered for publication, the following points should be addressed:
- My biggest concern is that from Fig 2A-B, it looks as if NAC + Klotho induces more double-stranded breaks than any group excluding slow freezing with PBS. What is the explanation for this? What are the p values for NAC + Klotho group vs each of the other groups?
- The authors report that the number of primordial follicles are slightly lower in slow-frozen tissue than in the control mouse ovarian tissue. The data from supplementary figure 1 does not have any statistics analysis reported. Are these differences significant? If not, the authors cannot make this conclusion and have to rephrase this sentence to accurately reflect their results.
- For figure 1 (top), it will be useful for the authors to show arrows pointing towards the basal layers to make it easy to comprehend for readers no familiar with the histology of ovarian tissue. It was difficult for me to figure out what exactly I was supposed to focus on.
- For figure 4C, densitometry and statistics need to be done to supplement the gel images shown.
- The “n” number of replicates/biological repeats need to be stated clearly in the legend for each figure (and each panel).
- NAC should be introduced as an antioxidant in the introduction rather than in the discussion. Readers not familiar with NAC will wonder why it was used in all the cryopreservation experiments until they get to the discussion.
Reviewer 2 Report
Herein, the authors investigated the pre-treatment with either N-acetylcysteine (NAC) or klotho protein or their combination on slow-frozen mouse ovarian tissue to prevent freezing-induced oxidative stress. Based on their observations klotho protein was siggested as an effective antioxidant agent for cryopreservation of ovarian tissue.
Two major concerns arise from my review:
- authors should further discuss the biological significance of the use of the combination of the two compounds insted of klotho alone
- authors should clarify in the methods section which were the concentrations used for NAC, klotho and their combination. Did they perform a dose-response curve before choosing the optimal concentration?
Round 2
Reviewer 1 Report
The authors have satisfactorily addressed all my concerns. I recommend this manuscript for publication.
Author Response
Thank you for your review.